# Association between Soil Physicochemical Properties and Bacterial Community Structure in Diverse Forest Ecosystems

**DOI:** 10.3390/microorganisms12040728

**Published:** 2024-04-03

**Authors:** Bing Yang, Wanju Feng, Wenjia Zhou, Ke He, Zhisong Yang

**Affiliations:** 1Sichuan Academy of Giant Panda, Chengdu 610041, China; 19828320401@163.com (W.F.); wenjia851209@163.com (W.Z.); yangzhisong@126.com (Z.Y.); 2Key Laboratory of Southwest China Wildlife Resources Conservation (Ministry of Education), China West Normal University, Nanchong 637002, China; heke0611@163.com

**Keywords:** soil bacterial community, forest soil, vegetation type, amplicon sequencing, diversity, co-occurrence network, community assembly

## Abstract

Although the importance of the soil bacterial community for ecosystem functions has long been recognized, there is still a limited understanding of the associations between its community composition, structure, co-occurrence patterns, and soil physicochemical properties. The objectives of the present study were to explore the association between soil physicochemical properties and the composition, diversity, co-occurrence network topological features, and assembly mechanisms of the soil bacterial community. Four typical forest types from Liziping Nature Reserve, representing evergreen coniferous forest, deciduous coniferous forest, mixed conifer-broadleaf forest, and its secondary forest, were selected for this study. The soil bacterial community was analyzed using Illumina MiSeq sequencing of 16S rRNA genes. Nonmetric multidimensional scaling was used to illustrate the clustering of different samples based on Bray–Curtis distances. The associations between soil physicochemical properties and bacterial community structure were analyzed using the Mantel test. The interactions among bacterial taxa were visualized with a co-occurrence network, and the community assembly processes were quantified using the Beta Nearest Taxon Index (Beta-NTI). The dominant bacterial phyla across all forest soils were Proteobacteria (45.17%), Acidobacteria (21.73%), Actinobacteria (8.75%), and Chloroflexi (5.06%). Chao1 estimator of richness, observed ASVs, faith-phylogenetic diversity (faith-PD) index, and community composition were distinguishing features of the examined four forest types. The first two principal components of redundancy analysis explained 41.33% of the variation in the soil bacterial community, with total soil organic carbon, soil moisture, pH, total nitrogen, carbon/nitrogen (C/N), carbon/phosphorous (C/P), and nitrogen/phosphorous (N/P) being the main soil physicochemical properties shaping soil bacterial communities. The co-occurrence network structure in the mixed forest was more complex compared to that in pure forests. The *Beta*-NTI indicated that the bacterial community assembly of the four examined forest types was collaboratively influenced by deterministic and stochastic ecological processes.

## 1. Introduction

Forest ecosystems play a vital role in supporting global biodiversity and contributing to various ecosystem services that are essential for human well-being and the health of our planet. Bacteria, the most abundant and diverse microorganisms in the soil [1,2,3,4], participate in ecological processes and provide ecosystem services [1,5,6]. Bacterial communities in forest soils, driven by various ecological interactions and feedback loops, contribute significantly to nutrient cycling, organic matter decomposition, and plant health. Additionally, soil bacterial communities are sensitive to environmental changes. Numerous studies have shown that soil bacterial communities on a local scale change with vegetation types and succession stages [7,8] and soil prosperities, such as soil water content (SWC) [9,10], pH [9,11,12,13], soil organic carbon (SOC) [9,14], total phosphorous (TP), total nitrogen (TN), and available phosphorous (AP) concentrations [15], and C/N ratio [16]. However, due to high forest soil heterogeneity and complicated plant-soil interactions, the environmental factors driving the abundance, diversity, and composition of the soil bacterial community remain unclear. This greatly hampers our predictions about the responses of the soil bacterial community to global climate change. Under the Paris agreement on climate change reached in 2015, China promised to halt the rise of its carbon emissions by 2030 and pledge “carbon neutrality” by 2060. Since the soil bacterial community plays a crucial role in organic matter mineralization, nutrient cycling, plant production, and greenhouse gas emissions [17,18], a deep understanding of the associations among forest type, soil bacterial community, and related functions is of utmost importance.

Over the past few decades, advances in high-throughput sequencing technologies and computational methods have revolutionized the study of soil microbial communities, allowing researchers to delve deeper into the complex interactions and dynamics that govern their assembly. High-throughput sequencing of 16S rDNA in environmental samples enables assessment of bacterial beta diversity and determinants of its biogeography [12] and has been applied to assess the diversity and functions of bacterial communities in forests [4,19,20,21,22,23]. The development of statistical methods makes it possible to gain a comprehensive understanding of the responses of the soil bacterial community to environmental change. For instance, the co-occurrence network analysis can offer new insights into the putative interactions between taxa in complex microbial communities [9,20,24,25]. Besides, it has proven powerful in identifying keystone taxa that have the largest influence in a community [20,25,26,27]. To date, co-occurrence analysis has been widely applied to exploring the responses of the microbial community to environmental changes in forest soils [20,21,23,28]. Additionally, community assembly offers a framework to explore the mechanisms driving microbial community dynamics. Community assembly theories, such as the niche theory [29] and the neutral theory [30], have provided conceptual frameworks to elucidate the underlying mechanisms driving the composition and structure of soil microbial communities [31,32]. Various studies have suggested that soil bacterial community assembly is combined and driven by deterministic and stochastic processes. However, there is no consensus yet on the relative importance of deterministic and stochastic processes in shaping soil bacterial community assembly, particularly in forest ecosystems.

The original intention of establishing Giant Panda National Park was to preserve the giant panda (*Ailuropoda melanoleuca*) and its habitat and to restore and enhance ecosystem multifunctionality. Studies have demonstrated that giant pandas’ habit are found in primary forests [33] and in secondary forests [34], whereas giant pandas are more abundant in traces in primary forests in comparison to secondary forests [35]. The soil bacterial community is a critical participant in the ecological process and thus an important resource to utilize in ecological restoration. However, its diversity and driving force in giant panda habitat remain poorly understood, greatly restricting our manipulation of the microbiome to improve soil health and enhance ecosystem multifunctionality.

The objectives of the present study are to explore the association between forest stand-type soil physicochemical properties and the composition, diversity, co-occurrence network topological features, and assembly mechanisms of the soil bacterial community. We hypothesize that (1) forest type significantly influences composition, diversity, co-occurrence topological features, and community assembly; (2) soil properties directly shape soil bacterial community features and co-occurrence pattern; and (3) ecological community assembly processes of soil bacterial communities are dominated by stochastic processes.

## 2. Material and Method

### 2.1. Site Description

The Liziping National Nature Reserve is a core distribution area and critical habitat for the giant panda. This reserve encompasses 47,885 hm^2^ of rugged ridges and narrow valleys at elevations ranging from 1330 to 4550 m. As reported, the mean annual temperature and precipitation are 11.7–14.4 °C and 800–1250 mm, respectively. The soil type of this area is braunerde, or brown forest soil.

### 2.2. Soil Sampling

Soil samples were acquired in October 2020 from the four selected forest types, namely coniferous (Con) forest, evergreen broadleaf (Bro) forest, secondary (Sec) forest, natural deciduous broadleaved and evergreen broadleaved mixed (Mix) forest. Within each forest type, nine sampling plots were established; dominant tree species, canopy cover, and dominant understory species were considered among the three plots to avoid unrepresentative areas. Within each plot, three 5 m × 5 m sized quadrates were established, and soil samples (0–20 cm) were collected using a 5 cm diameter soil auger from 10 locations along an S-shaped path after removing superficial debris (e.g., leaves, dry vegetation, and litter), mixed uniformly, and pooled to ensure that the soil samples were representative. The soil samples were then transferred to sterilized press seal bags and stored in an icebox. Next, the samples were taken to the laboratory and immediately sieved to remove stones and plant materials using a 2 mm mesh. One portion of the samples was air-dried and used to determine soil physicochemical properties, whereas the remainder was kept at −80 °C before soil bacterial community analysis.

### 2.3. Soil DNA Isolation and PCR Amplification

DNA extraction was carried out on 0.5 g frozen bulk soil samples using the E.Z.N.A.^®^ soil DNA Kit (OMEGA Bio-Tek, Norcross, GA, USA) following the provided instructions. Subsequently, the concentration and purity of the extracted DNA were determined through the utilization of a NanoDrop 2000 UV-Vis spectrophotometer (Thermo Scientific, Wilmington, NC, USA) and 1% agarose gel. The target sequence for analysis was the 16s rRNA V3-V4 region of bacteria, for which PCR amplification was conducted using specific primer pairs 338F (5′-ACTCCTACGGGAGGCAGCAG-3′) and 806R (5′-GGACTACHVGGGTWTCTAAT-3′) on an ABI GeneAmp^®^ 9700 PCR thermocycler (ABI, Los Angeles, CA, USA). The PCR mixtures were meticulously prepared, comprising various components such as high-fidelity DNA polymerase, reaction buffer, dNTPs, DNA templates, and specific primers. The amplification process involved a series of cycles with specified temperature and duration parameters, ultimately leading to the desired amplification of the target sequence in the extracted DNA. The PCR mixtures were prepared using 0.25 μL of Q5 high-fidelity DNA polymerase, 5 μL of reaction buffer (5×), 5 μL of high GC buffer (5×), 2 μL of dNTPs (10 mM), 2 μL of DNA templates, 1 μL each of forward (10 μm) and reverse (10 μm) primers, and 8.75 μL of ddH2O PCR. The specified amplification conditions, including denaturation, annealing, and extension steps, were diligently followed to facilitate the amplification of the target sequence. The amplification conditions were an initial denaturation at 95 °C for 3 min, followed by 27 cycles of denaturation at 95 °C for 30 s, annealing at 55 °C for 30 s, and extension at 72 °C for 45 s, with a final extension at 72 °C for 10 min.

### 2.4. Illumina Miseq Sequencing

The PCR amplification product was extracted using a 2% agarose gel and purified using an AxyPrep DNA Gel Extraction Kit (Axygen Biosciences, Union City, CA, USA). Purified amplicons were pooled in equimolar amounts and paired-end sequenced on an Illumina Miseq PE 300 platform (Illumina, San Diego, CA, USA) according to the standard protocols of Shanghai Personalbio Technology Co., Ltd. (Shanghai, China). The raw reads for all samples were sent to the NCBI Sequence Read Archive (SRA) database (accession number: PRJNA1005035).

### 2.5. Bioinformatic Analyses

The analysis of raw sequences was carried out using the QIIME2 2019.4 platform [36], where the sequences were trimmed, quality filtered, denoised, merged, and chimeras removed with the DADA2 pipeline [37]. Subsequently, sequences with a 100% match were grouped into amplicon sequence variants (ASVs) [37,38]. ASVs that were not singleton were aligned using MAFFT [39], and a phylogenetic tree was constructed using fasttree2 [40]. Following this, diversity metrics such as observed species, Chao1, Shannon, Simpson, and Faith’s PD were computed using QIIME2 [41].

### 2.6. Soil Physicochemical Parameters Measurements

Soil physicochemical properties were assayed with a standardized method. Specifically, SWC was determined with ten grams of field-moist soil samples at 105 °C for 24 h. Soil pH was measured using a glass electrode pH meter (the soil-to-water ratio was 1:2.5). The SOC content was measured using the high-temperature external heat dichromate oxidation capacity method. TN and AN were determined using the Kjeldahl digestion method and the alkaline hydrolysis method, respectively. TP was measured using the Mo-Sb anti-spectrophotometric method. Available phosphorus was extracted using an HCl-NH4F solution and determined using the molybdenum-antimony resistance colorimetric method. Soil C/N, C/P, and N/P were calculated based on SOC, TN, and TP.

### 2.7. Statistical Analyses

The effects of forest type on soil physicochemical parameters and bacterial community diversity were examined using one-way ANOVAs or non-parametric tests, depending on the normality and homogeneity of the variance. If a significant effect was detected, the differences were further assessed using the TukeyHSD test or pairwise. *t* test with the “Benjaminiand-Hochberg” method to adjust the *p* value. Spearman’s correlation analysis was utilized to assess the relationship between bacterial alpha diversity and soil physicochemical properties in R studio.

The effects of forest type on soil bacterial community composition and structure were examined using the following protocol: initially, Venn diagrams were generated to visualize the unique and shared amplicon sequence variants (ASVs) among the samples in R v4.2.1 using the “Venn Diagram” package. Additionally, heatmaps of the top 50 genera per sample were created in R using the “ggplot” and “pheatmap” packages. Furthermore, linear discriminant analysis (LDA) effect size (LEfSe) analysis was conducted to identify bacterial indicator taxa based on a normalized relative abundance matrix across groups using default parameters. Nonmetric multidimensional scaling (NMDS) based on Bray–Curtis matrices was performed to characterize the composition of the soil bacterial community using the “vegan” package. Analysis of similarities (ANOSIM) and permutation multivariate analysis of variance (PERMANOVA) were carried out to assess the statistically significant differences among bacterial communities using Bray–Curtis distances and 999 permutations. The Bray–Curtis distance was chosen due to its popularity as a similarity index for abundance data [16]. Redundancy analysis (RDA) based on Hellinger distance, a commonly used method for quantifying the similarity between two probability distributions [16], and the Mantel test with 999 permutations were employed to determine the effects of soil properties on the bacterial communities using the “vegan” and “ade4” packages in R studio.

The co-occurrence patterns of soil bacterial communities among the different forest types were determined using network analysis based on correlations of the 100 most abundant ASVs in the soil bacterial community. Co-occurring networks based on Spearman correlation analysis in this study were conducted using the “psych” package in R studio. The co-occurrence patterns of soil bacterial communities were studied based on strong correlations (*r* > 0.6) and significant correlations (*p* < 0.01). Network analysis and visualization were performed using igraph “package” [42].

The assemblage mechanisms of soil bacterial communities were analyzed through combined quantifying niche width, Sloan neutral community model [43] and null model [44]. Niche width was measured according to Levins’ coefficient:Bi=1∑j=1rPij2
where *B_i_* is the habitat niche width of ASV_i_, and *P_ij_* is the proportion of ASV_i_ in the total ASVs within a given forest type *j*. The average number of all ASVs’ B was calculated to represent the niche width of the soil bacterial community. The neutral community model was fitted by the nonlinear least-square fitting method, and the 95% confidence interval was predicted by the “Hmisc” package [45]. In terms of the null model, Beta Taxon Index (βNTI) and Raup–Crick (RC-Bray) were calculated to represent phylogenetic and taxonomic diversity [45,46,47]. |βNTI| > 2 indicates the dominance of deterministic processes, while |βNTI| < 2 indicates the dominance of stochastic processes. |βNTI| < 2 and RCBray < 0.95 represent homogenizing dispersal. |βNTI| < 2 and RCBray > 0.95 represent dispersal limitations. |βNTI| < 2 and |RCBray| < 0.95 represent “undominated” assembly (mainly consists of weak selection, weak dispersal, diversification, and/or drift). βNTI < 2 represents homogeneous selection. βNTI > 2 represents variable selection.

## 3. Results

### 3.1. Soil Physicochemical Properties

The soil moisture, SOC, TN, AN, TP, C:P, and N:P ratios varied greatly depending on forest type (Figure 1). The Bro Forest had the highest SWC, TN, AN, C/P, and N/P. The Con Forest had the lowest SWC, TN, AN, C/P, and N/P, but the highest TP.

### 3.2. Composition and Diversity

The full dataset contained 3,924,845 sequences from 36 successfully amplified soil samples. After processing within Qiime 2, a total of 3,288,127 sequences and 31,340 ASVs were retained for further analysis. According to the Venn diagram of the soil bacterial community, the total number of ASVs of soil bacteria shared among the four forest types was 2222 (Figure 2), accounting for approximately 7.09% of all ASVs. The Sec Forest exhibited the largest number of unique ASVs, with 6428 specific ASVs accounting for 20.51% of the total ASVs. The largest number of shared soil bacterial ASVs was found in the pair of Sec and Con forests. There were 1395 shared soil bacterial ASVs between these two forests, accounting for approximately 4.45% of the total ASVs. The dominant bacterial phyla across all forest soil samples were Proteobacteria (relative abundance 45.17%), Acidobacteria (21.73%), Actinobacteria (8.75%), and Chloroflexi (5.06%), whereas Bacteroidetes, Planctomycetes, Verrucomicrobia, Thaumarchaeota, Gemmatimonadetes, Rokubacteria, and other bacterial species occupied only a minor fraction of the bacterial community composition in bulk soils at the topsoil layer (Figure 3). Further analysis showed that the family with the greatest relative abundance of sequences was the Gammaproteobacteria (Phylum: Proteobacteria), and different taxa showed distinct associations with the same soil physiochemical property (Table A1 in Appendix A). Forest type significantly affected the Chao1 estimator of richness, observed ASVs, and faith-PD index (Figure 4). Specifically, the Chao1 estimator of richness, observed ASVs, and faith-PD index of the soil bacterial community in Sec Forest were higher than those in the Mix, Bro, and Con forests. Further analysis showed they were positively related to TN, AN, and AP contents, whereas they were negatively correlated with soil pH (Figure 4; Table A2 in Appendix A).

The NMDS ordination (Figure 5) and similarity analysis (Table 1) showed that bacterial community structures in bulk soils at the topsoil layer differed significantly across forest types.

RDA indicated that the soil bacterial community structure was significantly correlated with soil physicochemical properties (Figure 6, Table A3 in Appendix A). At the phylum level, the first two principal components of RDA explained 41.33% of the variations in the soil bacterial community (Figure 6). Envfit analysis showed that SWC, pH, SOC, TN, C/N, N/P, and C/P were significantly associated with RDA1 and RDA2 (Table A2 in Appendix A).

### 3.3. Co-Occurrence Network Topological Index of the Soil Bacterial Community

In general, all co-occurrence networks of soil bacterial communities were dominated by positive edges. Additionally, the network topological features of the four examined forest types varied greatly (Table 2; Figure 7; Figure A4; and Table A3 in Appendix A). Soil bacterial community co-occurrence networks in Mix and Sec forests are more complex and stable than those in Bro and Con forests. As observed, the number of nodes and edges are higher in Mix forests compared to pure forests, regardless of vegetation function type. Additionally, positive links in soil bacterial networks were greater than negative links (Table 2). In the networks, a total of 2, 4, 17, and 18 ASVs were keystones in Bro, Con, Sec, and Mix forests, respectively (Figure A4; Table A3 in Appendix A).

### 3.4. Assembly of the Soil Bacterial Community

As presented, the niche width (2.92–8.85) of the soil bacterial community varied greatly across forest types. Specifically, the niche width of the soil bacterial community in Con was significantly discrete than that in the Bro, Mix, and Sec forests (Figure 8a). The βNTI of soil bacterial communities were between 2 and −2 (Figure 8b), and dispersal-limited ecological process predominated in community assembly (Figure 8c). The neutral model fitted well for all soil bacterial communities in Bro, Con, Sec, and Mix forests. In addition, it successfully allowed the estimation of the relationship between the frequency of ASVs and their relative abundance, yielding R^2^ values ranging from 0.53 to 0.698 (Figure 8d).

## 4. Discussion

### 4.1. Dominant Taxa

In the present study, Proteobacteria, Acidobacteriota, Actinobacteria, and Chloroflexi were the dominant phyla across forest types (Figure 3). The dominance of these phyla has also been observed in forest soils in different regions. In Changbai Nature Reserve of China, Acidobacteria was found to be dominant in mixed conifer-broadleaf forest, while Proteobacteria showed a higher proportion in the coniferous forest ecosystem [2]. In the Barren Hills of North China, Proteobacteria, Acidobacteria, and Actinobacteria were the dominant phyla [48]. In the Qinling Mountains, the dominant bacterial phyla in secondary forest soils were Proteobacteria, Acidobacteria, Firmicutes, and Verrucomicrobia [8]. The prevailing dominance of these phyla implies their strong adaptability and critical ecological functioning in the forest ecosystem. For instance, Proteobacteria can thrive in various soil conditions, exhibit morphological, physiological, and metabolic diversity, and be involved in the degradation of soil organic matter and the biogeochemical cycling of carbon, nitrogen, and sulfur [49]. Acidobacteriota play a crucial role in maintaining soil health and nutrient cycling and are responsible for the degradation of plant- and microorganism-based polysaccharides. Their abundance is strongly correlated to soil pH [50,51,52,53] and nitrogen content [51,54]. Actinobacteria play an important role as symbionts and as pathogens in plant-associated microbial communities [55], and their abundance is positively correlated with soil organic matter and pH [56,57,58]. Their presence helps maintain soil fertility and health. The phylum Chloroflexi participates in hydrolyzing polysaccharides such as cellulose, xylan, and chitin [58]. The phylum Chloroflexi participates in hydrolyzing polysaccharides such as cellulose, xylan, and chitin [58]. Their presence in forest soils plays a critical role in nutrient availability and ecosystem productivity.

### 4.2. Soil Bacterial Community Diversity and Similarity

Our results, combined with previous studies, demonstrate that forest vegetation types select specific bacterial communities [2,59,60,61]. We observed a higher Chao1 index, observed ASVs, and faith-PD index of the soil bacterial community in Sec Forest compared to other forest vegetation types (Figure 2). Moreover, they were positively correlated with TN, AN, and AP contents, whereas they were negatively correlated with soil pH (Figure 2; Table A1 in Appendix A). The difference in soil bacterial community diversity and similarity between the Con and Bro forests is not surprising and can be attributed to several factors, including differences in carbon and nitrogen availability and root exudates. Con forest soils are also low in nutrient availability, which favors slow-growing bacterial species that are adapted to nutrient-poor environments. Previous studies have shown divergent correlations between soil bacterial richness and soil physicochemical properties. For instance, soil bacterial richness is correlated with soil C and N contents as well as the C/N ratio [62]. Additionally, bacterial alpha diversity is largely influenced by SOC and TN contents [63,64]. Our findings, combined with those of other studies, suggest that SOC and TN contents were not significantly correlated with bacterial diversity indices [13,65]. Contrary to the reported negative correlations between soil bacterial diversity and soil C/N [46,66], no evident correlation was found between diversity index and soil C/N in the present study.

In agreement with previous studies [60,61], our study identified significant differences in the bacterial community composition across Bro forests, Mix forests, and Con forests. The most likely explanation would be the differences in amount and quality of litter and root exudates. However, it is also worth noting that the observed dissimilarity in soil bacterial community across forest types can be explained by differences in soil moisture, pH, SOC, TN, C/N, C/P, and N/P (Figure 6, Table A3 in Appendix A), implying intricate associations between the soil microbial community and soil physicochemical properties. Similar findings have been reported elsewhere. Soil pH is a critical factor that affects the structure of the soil bacterial community [67,68,69,70,71]. Soil pH, TC, TN, AP, and AK were found to be the main abiotic factors structuring the bacterial communities in the forest soils of the Baishilazi Nature Reserve, China [65]. Similarly, soil pH, SOC, and TN were found to be the most important factors affecting the bacterial community in Purple Mountain Park forest soils in China [60]. Soil pH, SOC, AP, and ammonia nitrogen have been shown to be the best indicators for explaining the changes in the soil bacterial community in the Barren Hills of North China. Furthermore, soil pH and AP were found to be the main determinants of the soil bacterial community [72]. Additionally, aluminum, nitrogen, calcium, nutrient availability, and pH also contribute to the dissimilarity in the soil bacterial community in beech forests [73]. Finally, soil resource elemental stoichiometry was also found to play an essential role in bacterial diversity and community composition [9,18,74,75].

### 4.3. Co-Occurrence Network

The most noticeable finding of the present study was that the co-occurrence networks of soil bacterial communities in Mix and Sec forests are more complex and stable than those in Bro and Con forests (Figure 7). Firstly, the number of nodes and edges is higher in Mix forests compared to pure forests, regardless of vegetation function type. As is known to us, in a microbial co-occurrence network, nodes and edges indicate microbes and statistically significant associations between nodes, respectively; modules may represent different niches and ecological processes driving community structure [76,77]. Nodes with high betweenness centrality scores are crucial for maintaining network stability and structure [77]. A high betweenness centrality indicates that more microbes bridge connections between modules in the sub-networks; edge overlap among sub-networks indicates a similarity in the microbial co-occurrence patterns among these environments [77]. Negative edges suggest varying intensities of competition or niche differentiation in different environments. Low proportions of negative edges in sub-networks suggest a prevalence of collaboration or niche sharing [77]. Higher clustering coefficients and modularity values indicate that ecosystems have fragmented niches for the environmental adaptation of soil microorganisms [66,78]. Additionally, positive interactions reflect cooperation and niche differentiation among species in the networks, whereas negative interactions indicate competition and niche overlap among species [77,79]. An increase in positive associations implies that the co-occurrence network between synergistic groups was enhanced [66,80,81]. We observed that there were more positive links than negative links in the soil bacterial networks. This finding is consistent with previous studies [82,83]. The likely reasons why positive linkages might outweigh negative linkages among soil bacterial taxa are as follows: (1) the presence of certain bacteria promotes the growth or activity of others, leading to the formation of cooperative relationships; (2) forest soils are rich and diverse in organic matter, providing a wide range of substrates for bacterial growth, and different bacterial taxa specialize in utilizing specific organic compounds, leading to a complementary utilization of resources and enhanced overall ecosystem functioning; (3) some bacterial taxa may create favorable microenvironments for other species to thrive. For instance, one species might produce compounds that enhance the growth of neighboring species, leading to a positive feedback loop; (4) groups of bacteria work together to perform complex metabolic processes; and (5) the sampling bias or experimental design may result in certain taxa being more easily detected or studied, leading to an apparent dominance of positive interactions. Further studies are needed to elucidate the underlying mechanism of the prevalence of positive linkages among soil bacterial taxa in the co-occurrence network of the soil bacterial community in forest soils.

### 4.4. Bacterial Community Assembly Process

Identifying the relative contribution of deterministic and stochastic processes in soil microbial community assembly is a focal point in soil ecology. Deterministic processes, including environmental selection, species interactions, and niche differentiation, underpin the roles of biotic and abiotic filtering, leading to significant variation in community composition under different environmental conditions. Stochastic processes, such as dispersal limitation, undominated dispersal, and homogenizing dispersal, highlight the contribution of probabilistic dispersal and ecological drift to community composition patterns [46,71]. Recent evidence also supports that life-history traits are important determinants governing soil microbial community assembly in terrestrial ecosystems [53,84]. The intricate network of soil bacterial communities is a fundamental component of forest ecosystems, playing a crucial role in nutrient cycling, organic matter decomposition, and overall ecosystem functioning. Understanding the assembly processes that govern the structure and composition of soil bacterial communities in forest ecosystems is crucial for devising effective strategies for sustainable forest management, conservation, and the preservation of essential ecosystem services.

Previous studies suggest that soil bacterial community assembly is driven by a combination of deterministic and stochastic processes [8,46], but the relative importance of these two processes varies greatly depending on forest stand age [8], environmental factors [53,85,86,87], plant diversity [8], SOC [85,88], soil pH [86,89], soil moisture [87,89], and soil fungi [90] as critical mediators. Numerous pieces of evidence support that the soil bacterial community is primarily dominated by stochastic processes [53,71,84,91]. In contrast, the findings of the present study highlight the dominance of deterministic processes. The likely reasons are as follows: Firstly, we did not account for the potential effects of vegetation composition and stand age on the soil bacterial community. In previous studies, vegetation, associated soil physicochemical properties, and stand age have been shown to influence bacterial community composition [70,72,81]. Additionally, we did not consider the effects of altitude on the soil bacterial community, despite the pronounced altitudinal patterns observed in soil bacterial communities [70,89,92,93]. Finally, the observed patterns are based on a single sampling event. However, the effects of forest vegetation type on the soil bacterial community were seasonal-dependent [94], and the assembly processes of soil bacterial communities are dynamic and context-dependent [8,31,87]. Despite these limitations, the present study demonstrates that forest type is an important factor shaping the soil bacterial community, and its community assembly processes were shaped by a combination of deterministic and stochastic ecological processes.

## 5. Conclusions and Implications

In conclusion, the findings provide solid empirical evidence of close interlinkages among forest vegetation type, soil physicochemical properties, and the soil bacterial community, supporting our first and second hypotheses. Furthermore, Mix and Sec forests foster complex and robust co-occurrence networks compared with Bro and Con forests, implying increased biodiversity can enhance ecosystem resilience and functioning. However, the soil bacterial community of the four examined forests is consistently primarily controlled by deterministic processes, leading to the rejection of our third hypothesis. These findings enhance the holistic understanding of the interlinkages among forest vegetation type, soil physicochemical properties, and the soil bacterial community, which is essential for effective forest management, biodiversity conservation, ecological restoration, and sustainable ecosystem functioning. Studying the relationships between forest vegetation type, soil properties, and soil bacteria allows forest managers to assess the overall health and functioning of the ecosystem. This information can guide conservation efforts and sustainable management practices. Managing these interlinkages effectively enables forest managers to promote biodiversity conservation, protect endangered species, and implement strategies to enhance soil health and productivity. Knowledge of the interlinkages can also guide ecological restoration efforts in degraded forest ecosystems. By restoring the balance between vegetation, soil properties, and soil bacteria; restoration projects can be more effective in rebuilding healthy and resilient ecosystems. Finally, managing these interlinkages can help mitigate the effects of climate change by promoting carbon storage in soils and reducing emissions, as soil bacteria contribute to carbon sequestration and greenhouse gas emissions.

## Figures and Tables

**Figure 1 microorganisms-12-00728-f001:**
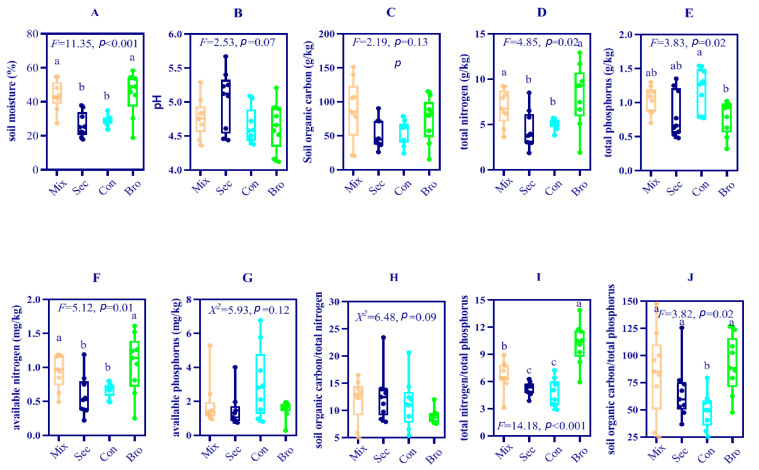
Soil physicochemical characteristics of the topsoil layer sampled from four selected forest types. (**A**) soil moisture, (**B**) pH, (**C**) soil organic carbon content, (**D**) total nitrogen content, (**E**) total phosphorus, (**F**) available nitrogen content, (**G**) available phosphorus content, (**H**) soil organic carbon/total nitrogen, (**I**) total nitrogen/total phosphorus, and (**J**) soil organic carbon/total phosphorus. Points represent the individual measures of the sequenced microbial communities, whereas the boxes span the interquartile range (IQR) of all measures, with the middle line being the median of the calculated indices. Different lowercase letters indicate a significant difference between forest types at *p* = 0.05. If a nonparametric test was used, the differences between groups were further examined using the Benjamini-Hochberg (BH) method to adjust the *p*-value. Otherwise, the differences between groups were further examined using the Tukey’s Honestly Significant Difference (HSD) test. The organ, blue, cyan and green represented natural deciduous broadleaved and evergreen broadleaved mixed forest, secondary (Sec) forest, evergreen broadleaf (Bro) forest and coniferous (Con) forest, respectively.

**Figure 2 microorganisms-12-00728-f002:**
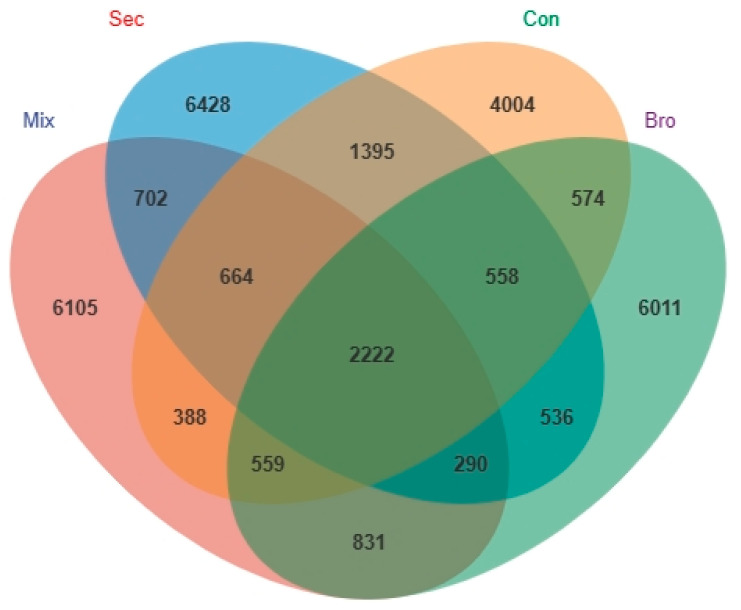
Venn diagram of the soil bacterial community of the evergreen broadleaved forest (Bro), coniferous forest (Con), natural deciduous broadleaved and evergreen broadleaved mixed forest (Mix), and secondary forest (Sec). The numbers within circles represent the specific bacterial amplicon sequence variants (ASVs) in those habitats, and the numbers in the overlaps represent the common bacterial ASVs between habitats. The core number represents the common bacterial ASVs present in all habitats.

**Figure 3 microorganisms-12-00728-f003:**
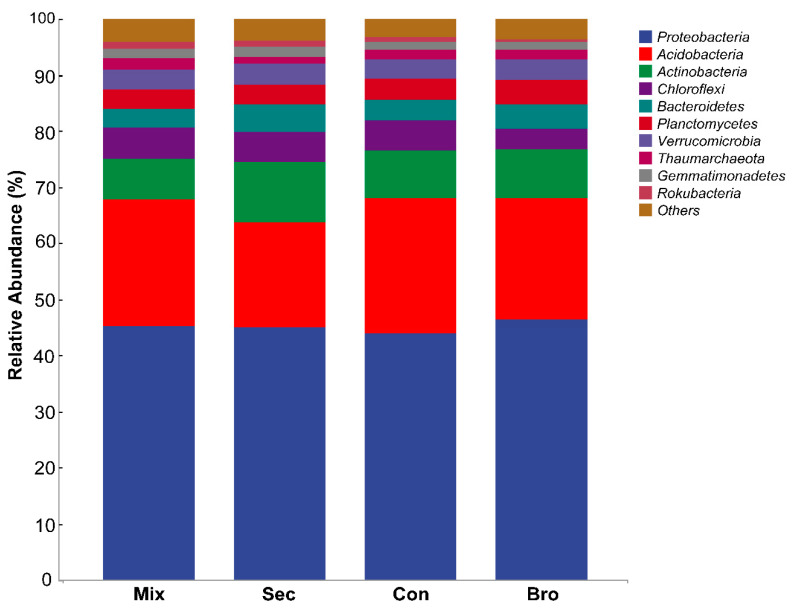
Relative abundances (%) of the main soil bacterial phyla. Low abundance phyla with less than 0.5% of the total sequences across all samples are grouped into “Others”. The colors of the bars represent the bacterial phyla, and the length represents the relative abundance of the bacterial phyla in the corresponding sample of evergreen broadleaved forest (Bro), coniferous forest (Con), natural deciduous broadleaved and evergreen broadleaved mixed forest (Mix), and secondary deciduous broadleaved and coniferous mixed forest (Sec).

**Figure 4 microorganisms-12-00728-f004:**
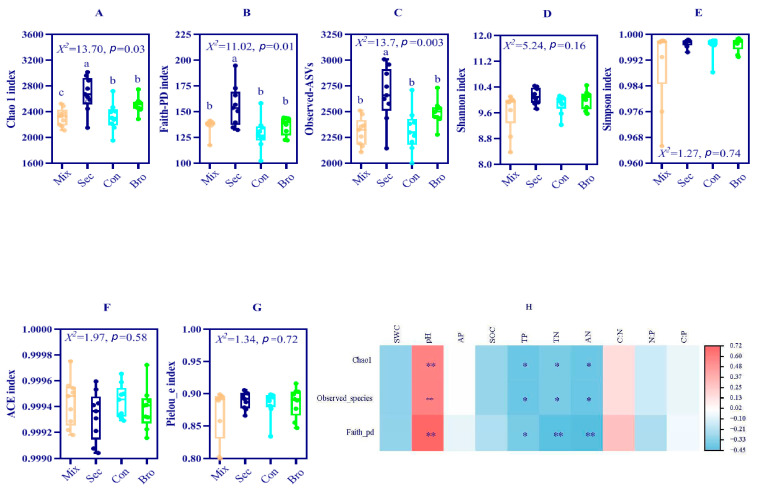
Bacterial richness, alpha diversity indices, and their correlations with the soil physicochemical properties of the topsoil layer sampled from four different forest types. Richness is estimated by the abundance-based coverage estimator (ACE), which is a nonparametric richness estimator based on the distribution of abundant (>10) and rare (<10) ASVs, and the Chao1 estimator of richness, which is a nonparametric richness estimator based on the distribution of singletons and doubletons. (**A**) Chao 1 index, (**B**) Faith-pd index, (**C**) observed-ASVs, (**D**) Shannon index, (**E**) Simpson index, (**F**) ACE index, (**G**) Pielou-e index, and (**H**) spearman correlations between soil bacterial community diversity and soil physicochemical properties. Different lowercase letters indicate a significant difference between forest types at *p* = 0.05. The organ, blue, cyan and green represented natural deciduous broadleaved and evergreen broadleaved mixed forest, secondary (Sec) forest, evergreen broadleaf (Bro) forest and coniferous (Con) forest, respectively. *, *p* ≤ 0.05; **, and *p* < 0.01.

**Figure 5 microorganisms-12-00728-f005:**
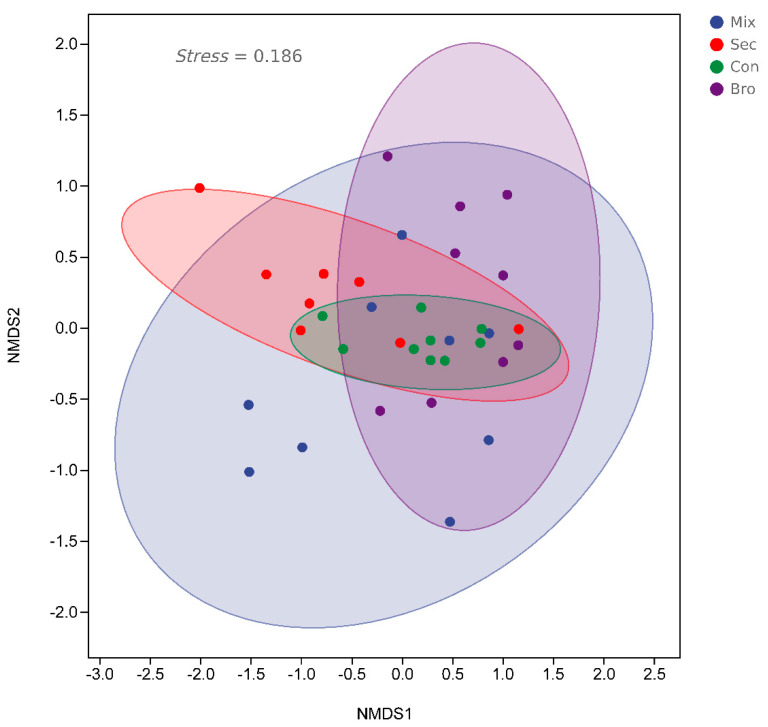
Comparisons of soil bacterial communities by non-metric multidimensional scaling (NMDS) plots based on Bray–Curtis similarity in evergreen broadleaved forest (Bro), coniferous forest (Con), natural deciduous broadleaved and evergreen broadleaved mixed forest (Mix), and secondary forest (Sec).

**Figure 6 microorganisms-12-00728-f006:**
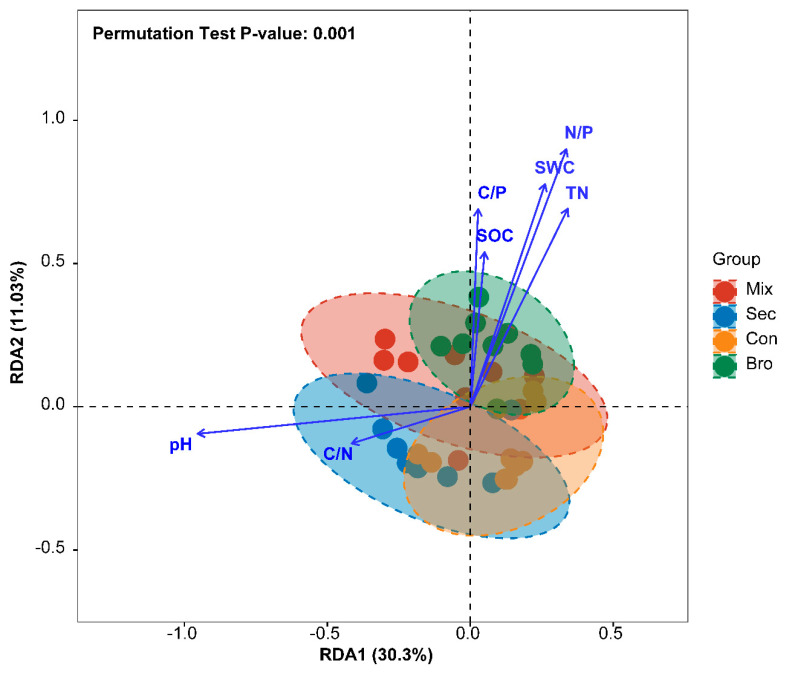
Biplot of site scores for the first two redundancy analysis (RDA) axes showing the associations between soil physicochemical properties and the Bray–Curtis distance of soil bacterial communities. The values of RDA axes 1 and 2 are the percentages explained by the corresponding axis. TN is total nitrogen, SWC is soil water content, SOC is total soil organic carbon, C/N is the ratio of soil total organic carbon to total nitrogen, C/P is the ratio of soil total organic carbon to total phosphorus, and N/P is the ratio of total nitrogen to total phosphorus. Bro is evergreen broadleaved forest, Con is coniferous forest, Mix is natural deciduous broadleaved and evergreen broadleaved mixed forest, and Sec is secondary forest.

**Figure 7 microorganisms-12-00728-f007:**
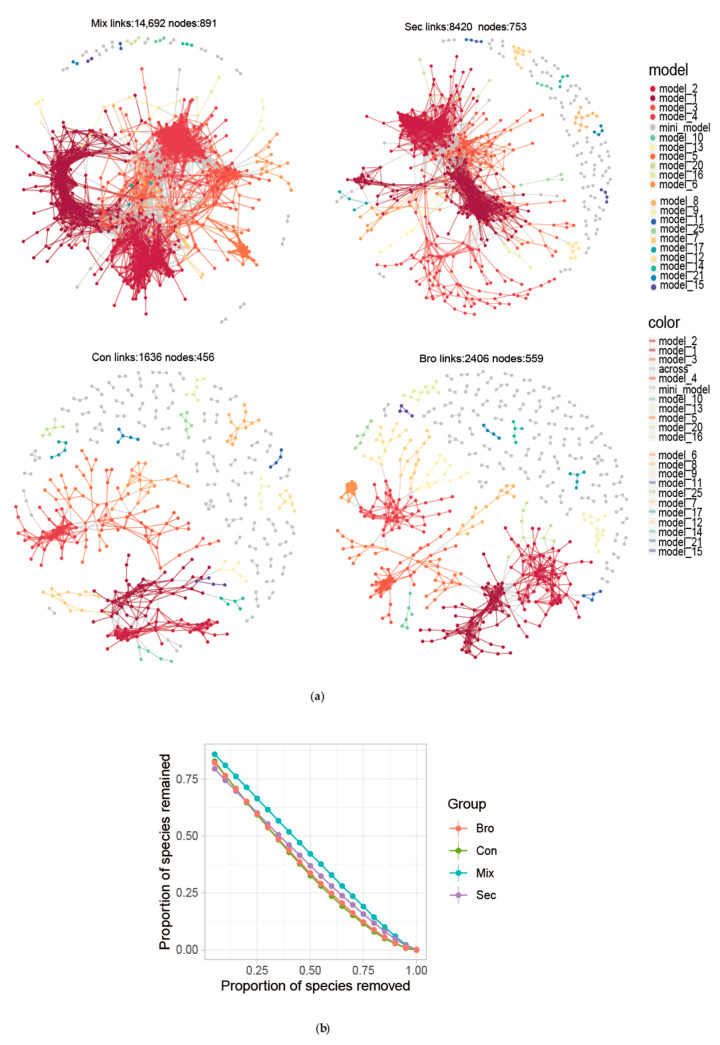
Co-occurrence network structure (**a**) and network robustness (**b**) of soil bacterial communities from evergreen broadleaved forest (Bro), coniferous forest (Con), natural deciduous broadleaved and evergreen broadleaved mixed forest (Mix), and secondary forest (Sec). Nodes are colored according to microbial phylum, and the nodes with a larger size show the potential keystone genera (correlation coefficient ≥ 0.6, *p* < 0.01). Circle and square node shapes represent bacteria and genera, respectively. Edges indicate correlations among nodes; the red and blue edges represent positive and negative correlations, respectively.

**Figure 8 microorganisms-12-00728-f008:**
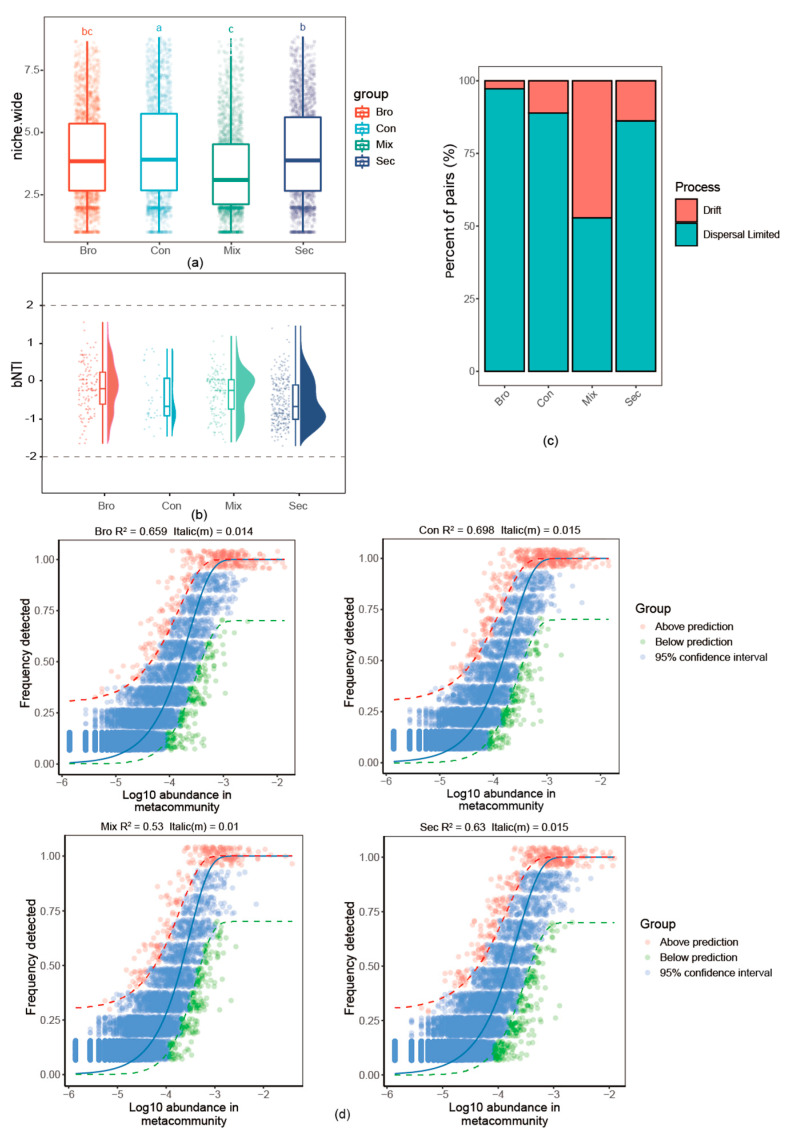
The niche width (**a**), the relative importance of different ecological processes based on the null model (**b**,**c**), and the relative abundance-frequency relationships (**d**) of the soil bacterial community under different forest types. Values assigned with the same letter were not significantly different by the post hoc Tukey honestly significant difference (HSD) test (*p* ≤ 0.05). The dashed blue line represents the 95% confidence interval above and below the prediction (the solid blue line). R^2^ indicates the coefficient of the neutral fit, and Nm indicates the meta-community size times immigration.

**Table 1 microorganisms-12-00728-t001:** The overall bacterial community structure comparison using three nonparametric statistical methods.

Object	Adonis			ANOSIM		MRPP		
	*F*	*R* ^2^	*Adjusted-p*	*R*	*Adjusted-p*	*A*	δ	*Adjusted-p*
Among Group	2.9559	0.2170	0.01	0.3364	0.01	0.0796	0.5696	0.001
Mix vs. Sec	2.5666	0.1382	0.012	0.2685	0.014	0.0489	0.6345	0.009
Mix vs. Con	1.2714	0.1195	0.023	0.2003	0.026	0.0387	0.5957	0.020
Mix vs. Bro	2.4631	0.1334	0.006	0.2857	0.008	0.0439	0.6295	0.008
Sec vs. Con	2.7965	0.1488	0.024	0.2850	0.019	0.0475	0.5641	0.022
Sec vs. Bro	4.8352	0.2321	0.006	0.6602	0.006	0.0977	0.6279	0.003
Con vs. Bro	3.0545	0.1603	0.006	0.3858	0.006	0.0581	0.5688	0.003

Note: Adonis, nonparametric multivariate analysis of variance (MANOVA) with the adonis function; ANOSIM, analysis of similarity; MRPP, multi-response permutation procedure. Bro, evergreen broadleaved forest; Con, coniferous forest; Mix, natural deciduous broadleaved and evergreen broadleaved mixed forest; Sec, secondary forest.

**Table 2 microorganisms-12-00728-t002:** Topological characteristics of the co-occurrence network of soil bacterial communities in soils with different dominant species.

Treatments	Nodes	Links	Positive Edges	Negative Edges	APL	ACC	Average Degree	Diameter	Density	Modularity
Mix	869	5464	4732	732	5.0237	0.4717	12.5754	12.9697	0.0145	2.4978
Sec	826	8420	5964	2095	5.1728	0.4679	14.4407	15.7853	0.0175	1.0741
Bro	551	737	554	183	6.8242	0.4208	13.4354	13.4354	0.0049	0.6006
Con	550	707	512	195	5.2440	0.4421	17.2239	17.2239	0.0047	0.4483

Note: Bro, evergreen broadleaved forest; Con, coniferous forest; Mix, natural deciduous broadleaved and evergreen broadleaved mixed forest; Sec, secondary deciduous broadleaved and coniferous mixed forest. ACC, average clustering coefficient; and APL, average path length.

## Data Availability

The datasets presented in this study can be found in online repositories.

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
