# Peer review of "Association between Soil Physicochemical Properties and Bacterial Community Structure in Diverse Forest Ecosystems"

_microorganisms, 2024, doi:10.3390/microorganisms12040728_

Round 1

Reviewer 1 Report

Comments and Suggestions for Authors

-          This paper correspond for scope of journal. 

-          The title corresponds to the content of the paper.  

-          This study represents a significant contribution to determining diversity of soil bacterial communities (plays a crucial role in organic matter mineralization, nutrient cycling, plant production and greenhouse gas emissions) in different forest type ecosystem to resolve functional interactions associations between its community composition, structure, co-occurrence patterns, and soil physicochemical properties in ecosystem of the mixed  and  forest pure forests (i.e. coniferous forest, evergreen broadleaf, secondary mixed forest, conifer–broadleaf, mixed forest) with focus in China.

-          The main question of paper addressed to establish role of association between soil physicochemical properties and the composition of bacterial communities as well their role in supporting biodiversity, and mitigation of climate change through increasing CO2 sequestration by root of forest plants as well as how the forest type is an important factor shaping soil bacterial community.

-          The aim of research  is  clearly and fully pointed  in abstract as well on the end of chapter of Introduction.

-          Key words are appropriate.

-          Scientific methodology is applied correctly for this type of study.

-          Results are clearly presented and discussed.

-          Tables, figures, pictures are clear.

-          The conclusions are clear but should be improved with more results data.

-          Manuscript is acceptable after minor revision!

Author Response

Point-by-point response to Reviewer #1

Q1: “This paper corresponds for scope of journal.

The title corresponds to the content of the paper. 

This study represents a significant contribution to determining diversity of soil bacterial communities (plays a crucial role in organic matter mineralization, nutrient cycling, plant production and greenhouse gas emissions) in different forest type ecosystem to resolve functional interactions associations between its community composition, structure, co-occurrence patterns, and soil physicochemical properties in ecosystem of the mixed and forest pure forests (i.e. coniferous forest, evergreen broadleaf, secondary mixed forest, conifer–broadleaf mixed forest) with focus in China.

The main question of paper addressed to establish role of association between soil physicochemical properties and the composition of bacterial communities as well their role in supporting biodiversity, and mitigation of climate change through increasing CO2 sequestration by root of forest plants as well as how the forest type is an important factor shaping soil bacterial community.

The aim of research is clearly and fully pointed in abstract as well on the end of chapter of Introduction.

Key words are appropriate.

Scientific methodology is applied correctly for this type of study.

Results are clearly presented and discussed.

Tables, figures, pictures are clear.”

A1: Thank you very much for reviewing our manuscript and giving us positive comments.

Q2: The conclusions are clear but should be improved with more results data.

A2: Thank you for this constructive suggestion. We have added necessary results. Please check the revision.

Q3: Manuscript is acceptable after minor revision!

A3: Thank you very much for the strong support to our work.

Reviewer 2 Report

Comments and Suggestions for Authors

In lines 43-47, the same text is repeated

Line 206 - reference should be added to (Dong et al., 2021)

The authors should carefully review the manuscript. There are some typing or grammatical errors.

line 170 write the full name for BH

In figures legends or other texts are very small or typed with fonts that make them impossible to understand.

Figure 8 spans 3 pages, which makes it difficult to read. Subfigures (a)-(c) seem to be too big, while subfigure (d) is too small. Maybe the authors should separate this figure.

Line 474 - repetition

In the text, the authors always use the full names of the four types of studied forests. In the figures, abbreviations are used. Sometimes it is awkward to read and compare text with figures. Also reading such long names repeated in the text seems to make it more difficult. Maybe the authors should define at the beginning of the manuscript common definitions of abbreviations and use them in the text.

On the contrary, In lines 217-220, the authors used abbreviations of soil parameters, while in described Figure 1 used full names. That also creates unnecessary difficulties in reading the manuscript. A similar difficulty is also present in other figures' descriptions.

In lines 217-220, the authors present results, which not always are confirmed by the statistical tests, as presented in the figure. That should be noted in the description of the results.

In Figure 4 the authors collected two types of results (a)-(g), and (h). These should be separated into two figures. Figure (h) is not clearly described in the caption.

In lines 275-277 describing Figure 5, the authors claim that there are four distinct clusters. In my opinion that claim is not supported by the presented results. There are some patterns visible in this figure, for example, showing that data for some forest types are more concentrated and for others more dispersed. But it is not equivalent to four clusters. Maybe it is just not written (or plotted) clearly.

Figure 7 is not clear, used font size is too small for reading.

Also, this figure is composed of two types of subfigures. In my opinion that should be separated into two figures or explained why they are grouped. The meaning of subfigure (b) is not explained in the figure caption.

In lines 334-337, the authors refer to Figure 8(a), that the niche width was found significantly broader for one of the types of forests. If I understand it correctly, that refers to the mean value. However, in Figure 8(a) position of the mean is not indicated. I assume that the horizontal line represents the median. 

The authors present in this figure that the statistical test confirmed the difference between "Con" and "Sec" forests. The plot shows that the distributions are very similar, at least when we look at statistics plotted as boxes. Maybe the mean values are different but are not indicated.

The meaning of boxes and other symbols in figures should be clearly explained. I can guess that the boxes span from 1st to 3rd quantile, but there are various standards in plotting this type of chart.

Why in Figure 8(a) some points are plotted outside of boxes? what is their meaning? Why only in this figure?

Why does Figure 8 group four different types of results, maybe it would be reasonable to split them into separate subfigures.

The convention is that Latin names should be typed using italic font.

The authors should carefully review the manuscript. There are some typing or grammatical errors.

In Section 2.7 the authors list used statistical methods, most of them in one long paragraph. Lines 172-190, but also in other paragraphs. It would help the readers if this enumeration was itemized, to more easy notice when starts description of each new method.

I also advise the authors to add a more description/interpretation of possible results of the methods of statistical analysis, and also references to papers describing them. That would be helpful to other researchers. Some of such information can be found in Introduction section.

Author Response

Point-by-point response to Reviewer #2

Q1: In lines 43-47, the same text is repeated

A1: Thank you very much for pointing out this error. We have deleted the repeated words.

Q2: Line 206 - reference should be added to (Dong et al., 2021)

A2: Done.

Q3: The authors should carefully review the manuscript. There are some typing or grammatical errors.

A3: We have tried our best to proofread and edited the manuscript thoroughly.

Q4: line 170 write the full name for BH

A4: Done.

Q5: In figures legends or other texts are very small or typed with fonts that make them impossible to understand.

A5: We strongly agree with your view. Figures, particularly figures 7 and 8 have been completely improved.

Q6: Figure 8 spans 3 pages, which makes it difficult to read. Subfigures (a)-(c) seem to be too big, while subfigure (d) is too small. Maybe the authors should separate this figure.

A6: We agree partly with your viewpoint, and we have improved and reconfigured these subfigures. However, we do not separate this figure because they combinedly display the relative importance of different ecological processes in soil bacterial community assemblage. We hope that your comments have been addressed accurately.

Q7: Line 474 - repetition

A7: We were sorry for this careless mistake, thank you for your reminder. We have deleted the repeated words.

Q8: In the text, the authors always use the full names of the four types of studied forests. In the figures, abbreviations are used. Sometimes it is awkward to read and compare text with figures. Also reading such long names repeated in the text seems to make it more difficult. Maybe the authors should define at the beginning of the manuscript common definitions of abbreviations and use them in the text. On the contrary, in lines 217-220, the authors used abbreviations of soil parameters, while in described Figure 1 used full names. That also creates unnecessary difficulties in reading the manuscript. A similar difficulty is also present in other figures' descriptions.

A8: We are greatly grateful for your professional comments. we

Q9: In lines 217-220, the authors present results, which not always are confirmed by the statistical tests, as presented in the figure. That should be noted in the description of the results.

A9:

Q10: In Figure 4 the authors collected two types of results (a)-(g), and (h). These should be separated into two figures. Figure (h) is not clearly described in the caption.

A10: We sincerely appreciate your careful reading and valuable comments. As you suggested, there were two types of results. In figures (a)-(g), the differences in richness and diversity indices, such as Chao 1 index, ACE index across four forest types were compared, whereas in figure g the correlations between soil physicochemical properties and diversity indices were showed. It seems that these are two separated results. However, these figures were used to show the differences and their key drivers of richness and diversity of soil bacterial community.

Q11: In lines 275-277 describing Figure 5, the authors claim that there are four distinct clusters. In my opinion that claim is not supported by the presented results. There are some patterns visible in this figure, for example, showing that data for some forest types are more concentrated and for others more dispersed. But it is not equivalent to four clusters. Maybe it is just not written (or plotted) clearly.

A11: Thanks a lot for your careful checks. We have re-written this part according to your suggestion.

Q12: Figure 7 is not clear, used font size is too small for reading. Also, this figure is composed of two types of subfigures. In my opinion that should be separated into two figures or explained why they are grouped. The meaning of subfigure (b) is not explained in the figure caption.

A12: We were greatly grateful for your reminder. We have changed the font size and added the explanation of subfigure (b) in the figure caption. However, we did not separate these subfigures into two figures because they showed the topological structure complexity and robustness of co-occurrence networks.

Q13: In lines 334-337, the authors refer to Figure 8(a), that the niche width was found significantly broader for one of the types of forests. If I understand it correctly, that refers to the mean value. However, in Figure 8(a) position of the mean is not indicated. I assume that the horizontal line represents the median. The authors present in this figure that the statistical test confirmed the difference between "Con" and "Sec" forests. The plot shows that the distributions are very similar, at least when we look at statistics plotted as boxes. Maybe the mean values are different but are not indicated. The meaning of boxes and other symbols in figures should be clearly explained. I can guess that the boxes span from 1st to 3rd quantile, but there are various standards in plotting this type of chart. Why in Figure 8(a) some points are plotted outside of boxes? what is their meaning? Why only in this figure?

A13: We were sorry to make you confused. May be this is because we used different charts. In figure 8(a), boxplot combined with jitter plot were used to show the differences in niche width between four forest types. In figure 8(b), boxplot combined with raincloud plot were used to show the differences in βNTI of soil bacterial communities between four forest types. However, the meaning of boxes is the same. As is clearly depicted in the figure below, these plots encompass the upper whisker, upper quartile, median, lower quartile and lower whisker and outliners.

Q14: Why does Figure 8 group four different types of results, maybe it would be reasonable to split them into separate subfigures.

A14: Although four distinct types of results were shown in Figure 8, they combined to show the different ecological processes in soil bacterial community assemblage. Since they demonstrated the same question, it is not necessary to split them.

Q15: The convention is that Latin names should be typed using italic font.

A15: We have consulted the requirement for Latin names and corrected their format according to the rule.

Q16: In Section 2.7 the authors list used statistical methods, most of them in one long paragraph. Lines 172-190, but also in other paragraphs. It would help the readers if this enumeration was itemized, to more easy notice when starts description of each new method. I also advise the authors to add a more description/interpretation of possible results of the methods of statistical analysis, and references to papers describing them. That would be helpful to other researchers. Some of such information can be found in Introduction section.

A16: Thank you for this constructive suggestion. We strongly agree with your views and make some changes. Please check the revision.